# Bayesian Regularized SEM: Current Capabilities and Constraints

Sara van Erp

Department of Methodology and Statistics, Utrecht University, 3508 TC Utrecht, The Netherlands;
s.j.vanerp@uu.nl

**Abstract:** An important challenge in statistical modeling is to balance how well our model explains the phenomenon under investigation with the parsimony of this explanation. In structural equation modeling (SEM), penalization approaches that add a penalty term to the estimation procedure have been proposed to achieve this balance. An alternative to the classical penalization approach is Bayesian regularized SEM in which the prior distribution serves as the penalty function. Many different shrinkage priors exist, enabling great flexibility in terms of shrinkage behavior. As a result, different types of shrinkage priors have been proposed for use in a wide variety of SEMs. However, the lack of a general framework and the technical details of these shrinkage methods can make it difficult for researchers outside the field of (Bayesian) regularized SEM to understand and apply these methods in their own work. Therefore, the aim of this paper is to provide an overview of Bayesian regularized SEM, with a focus on the types of SEMs in which Bayesian regularization has been applied as well as available software implementations. Through an empirical example, various open-source software packages for (Bayesian) regularized SEM are illustrated and all code is made available online to aid researchers in applying these methods. Finally, reviewing the current capabilities and constraints of Bayesian regularized SEM identifies several directions for future research.

**Keywords:** structural equation modeling; Bayesian; regularization; penalization; shrinkage prior

## 1. Introduction

*"All models are wrong, but some are useful."* George Box

When we try to describe real-world phenomena using statistical models, we need to strike a balance between how accurately we describe the phenomenon in question and how parsimonious our description is. On the continuum from restrictive to free models, we wish to find the sweet spot where our description captures the characteristics of the phenomenon we try to describe in the most parsimonious way possible. For example, suppose we wish to predict an outcome using a regression model. We might include hundreds or even thousands of predictor variables to obtain the very best prediction possible but by doing so, lose the advantage of a parsimonious explanation as to which factors are most important in predicting the outcome. Our aim should therefore be to find the main predictors that have a substantial influence on our outcome and ignore the rest. Within our statistical model, the regression coefficients for those main predictors should be estimated, while all other coefficients should be set to zero.

Similarly, in structural equation modeling (SEM), we want to balance freely estimating all parameters in the model with restricting some of them in order to obtain a parsimonious description of the phenomenon under investigation. If we freely estimated as many parameters as we have observed covariances, we would obtain the saturated model. Despite fitting perfectly to the data, this model would be useless since it does not provide a parsimonious description. On the other end of the continuum would be the most restrictive independence model that sets all parameters except the variable means and variances to zero, which is also useless since it offers no description of the phenomenon at all. The most parsimonious model that still provides a sufficiently accurate explanation for the

phenomenon under investigation lies somewhere between the saturated and independence model (see Figure 1). Note that what constitutes a "sufficiently accurate" explanation will vary between studies and researchers.

The traditional confirmatory approach within SEM uses theory to guide which paths are fixed and which are freely estimated. The resulting model will be somewhere on the continuum from restrictive (i.e., the independence model) to free (i.e., the saturated model; see Figure 1). Subsequently, the fit of the model is assessed through the likelihood-ratio test and approximate fit indices. If the fit is not acceptable, the model can be adapted by freeing parameters, for example, based on modification indices [1]. By doing so, the model is shifted further from restrictive to free on the continuum.

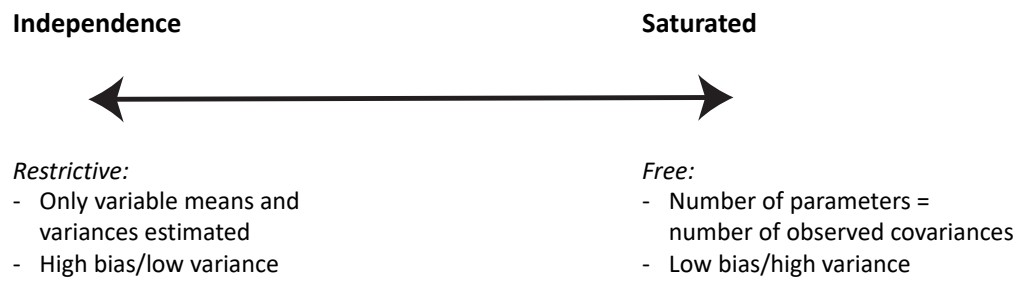

**Figure 1.** Models can be on a continuum from restrictive to free.

The main issue with this approach is that shifting the model towards the free end of the continuum until model fit is acceptable focuses mainly on reducing the bias of the model. However, there is a well-known tradeoff between the bias and the variance of a method. As we continue freeing parameters in our model, we might reduce bias, but at the cost of increased variance. In the end, using modification indices to guide the model-building process might lead to a model that performs very well on the current sample (due to the low bias) but does not generalize to other samples (due to the high variance). This is the problem of chance capitalization well known when using modification indices [2]. A more practical downside is that modification indices require a multi-step approach since the model needs to be refitted after each restriction has been freed until the fit of the model is acceptable.

In order to obtain models that automatically balance bias and variance, we can turn to methods that have been used with success to achieve this goal in the context of regression models. These penalization or regularization methods add a penalty term to the estimation procedure that serves as a failsafe to protect model parsimony and thereby avoid the problem that a model becomes too complex and results in high variance. Within the context of SEM, various regularization approaches have been proposed, for example by [3,4], with general implementations in R available through the R-packages `regsem` [5] and `lslx` [6].

By applying regularization methods in structural equation models, convergence problems can be solved and inadequate performance of test statistics due to small samples can be improved [7–10]. In exploratory factor models, regularized SEM has been used to produce a sparse loading matrix [11–14], and in confirmatory factor models, regularization can be used to achieve a simple structure in one step while allowing cross-loadings to be estimated [3]. Regularization can help determine which covariates predict a latent variable in multiple indicators multiple causes (MIMIC) models [3,15] and identify potential mediators in mediation models [16–18]. It has been applied in discrete latent variable models (i.e., latent class models) to provide an alternative to purely exploratory or confirmatory approaches that can provide interpretable and stable parameter estimates for relatively small samples [19,20]. In the context of comparing groups, regularization methods have been applied to detect violations of measurement invariance or differential item functioning [21–23]. Furthermore, regularization enables the estimation of both undirected and directed contem-

poraneous effects in vector autoregressive (VAR) models [24]. Finally, ref. [25] compared various regularization approaches in SEMs where cross-loadings and/or latent interactions were misspecified and found that these approaches resulted in less bias compared to standard maximum likelihood (ML) estimation. As a result, they argued that regularization methods might even be used as a default estimator in the context of SEM.

Classical regularized SEM approaches rely on the frequentist framework and add the penalty term to the traditional optimization problem, for example, the maximum likelihood cost function. An alternative way to penalize model parameters in SEM would be through the use of priors in a Bayesian SEM analysis. It is well known that results obtained with various classical penalties, such as the ridge and lasso penalty, are equivalent to specific posterior estimates given particular priors. Thus, instead of relying on a penalty function, Bayesian regularized SEM relies on the prior distribution to enforce regularization. Ref. [26] presents a comparison between the classical and Bayesian regularized SEM frameworks. The current work can be seen as an extension of [26] in which I focus primarily on the Bayesian regularized SEM framework and, as such, can cover it in more depth compared to [26].

There are various advantages to using Bayesian regularized SEM instead of the classical framework (see Section 3). Because of the wide range of possibilities when using Bayesian regularized SEM, for example, in terms of different models, priors, and estimation methods, there is not a single user-friendly, all-purpose software package available for Bayesian regularized SEM. Therefore, the goal of the current paper is to provide an overview of the available software packages. The focus is on Bayesian regularized SEM, but I include three open-source packages for classical regularized SEM as well. The illustration of these packages on an empirical example serves three goals: (1) to provide a tutorial on using the various existing software packages by making annotated R code available; (2) to provide guidance on various general principles that are important when using Bayesian regularized SEM such as choosing the shrinkage prior and scaling the data; and (3) to highlight current limitations of Bayesian regularized SEM, which will hopefully prompt other researchers to conduct future research in this area and make Bayesian regularized SEM even more useful and practical.

The outline of this paper is as follows. Section 2 explains classical regularized SEM, followed by a description of Bayesian regularized SEM in Section 3 including an extensive review of theoretical developments in this area. Section 4 presents an overview of available software packages for regularized SEM, with four open-source packages being illustrated on a realistic example in Section 5. Finally, the paper is concluded with a discussion and directions for future research in Section 6.

## 2. Regularized SEM

Regularization or penalization is a statistical technique that has gained popularity, especially in the context of regression models, to automatically select predictors while guarding against overfitting. Regularized regression approaches add a so-called "penalty term" to the minimization of the sum of squared residuals. The goal of this penalty term is to shrink small regression coefficients towards zero while leaving large coefficients large. The manner in which this is achieved differs between various penalty terms that can be used. For example, the least absolute shrinkage and selection operator (lasso; [27]) can result in small coefficients becoming exactly zero, whereas the ridge penalty [28] can only shrink them to be close to zero. The third most well-known penalty function, the elastic net [29] provides a combination of the lasso and ridge penalty functions. In addition, many more advanced penalty functions have been developed in order to achieve specific shrinkage behaviors (such as shrinking groups of coefficients simultaneously through the group lasso; [30]). A comprehensive introduction to and overview of various regularized regression methods in the classical framework is provided in [31].

Despite being developed within the regression framework, regularization approaches can be applied in other statistical models as well. Indeed, as SEM can be seen as a general-

ization of regression analysis, the application of regularization methods in SEM is similar to their application in regression. Specifically, a penalty function is added to the fitting function that is minimized, i.e.,

$$F_{regsem}(S, \Sigma(\theta)) = F(S, \Sigma(\theta)) + \lambda P(\theta_{reg}) \tag{1}$$

Here, $S$ is the observed sample covariance matrix and $\Sigma(\theta)$ is the model implied covariance matrix. $F(S, \Sigma(\theta))$ is the fitting function used, for example, the maximum likelihood function, and $P(\theta_{reg})$ is a penalty function that sums the values of the parameters to be regularized ($\theta_{reg}$). $\lambda$ is the regularization parameter that decides the amount with which parameters are shrunken towards zero: if $\lambda = 0$, regular SEM is performed with no shrinkage and if $\lambda = \infty$, regularized SEM is performed with all parameters included in the penalty function shrunken towards zero. See also [3] for a general overview of regularized SEM.

### 3. Bayesian Regularized SEM

In classical regularized SEM, a penalty function is added to the fitting function that is minimized. In Bayesian regularized SEM, a prior distribution is multiplied by the likelihood of the data to obtain the posterior distribution. I do not discuss the basics of Bayesian analysis or its differences with classical approaches here, but refer to introductory text books on this topic, such as [32]. Of importance within the current context are that: (1) the prior distribution can be seen as the Bayesian equivalent of the penalty function; and (2) instead of obtaining point estimates as final results, Bayesian regularized SEM leads to posterior distributions for each of the parameters.

Apart from uniform priors, all priors enforce some regularization of parameters, and so the question arises as to how to define Bayesian regularization to distinguish it from Bayesian estimation in general. For the purpose of this paper, I refer to Bayesian regularization whenever a prior distribution is used with the explicit goal of shrinking small parameters towards zero to avoid overfitting or to achieve model identification. Priors that are used specifically to achieve this goal are called shrinkage priors.

Many different shrinkage priors exist. Although all shrinkage priors are peaked around zero (to shrink small effects towards zero) and most shrinkage priors have heavy tails (to allow substantial effects to remain large), the exact shapes of the shrinkage priors differ, resulting in different shrinkage behaviors. Ref. [33] provides a comprehensive overview of popular shrinkage priors. Figure 2 visualizes the shrinkage behavior of a normal or ridge prior. The high peak at zero of the normal shrinkage prior pulls the posterior towards zero. The amount of shrinkage towards zero in the normal prior is determined by the variance or standard deviation of the distribution relative to the sample size of the data. For a fixed sample size, a smaller prior variance leads to a more peaked distribution and more shrinkage towards zero. Priors that have more prior mass on values away from zero (heavy tails) allow substantial effects to escape this shrinkage. The normal prior does not have heavy tails and therefore pulls all effects towards zero quite heavily, including substantial effects. This can introduce more bias compared to heavier-tailed shrinkage priors such as the horseshoe prior.

It is well known that the solutions obtained using various classical penalty functions can also be obtained in a Bayesian framework by specifying a particular prior distribution combined with a specific posterior point estimate. For example, ref. [28] mentions that the ridge estimate can be obtained as the posterior mean when the parameters are given a normal prior distribution and [27] points out how the lasso solution is equivalent to the posterior mode estimate when the parameters are given double-exponential priors (see [33] for more equivalencies between Bayesian and classical shrinkage methods). However, there are various differences between classical and Bayesian regularized SEM, with some differences being advantageous for classical SEM, while others are advantageous for Bayesian regularized SEM.

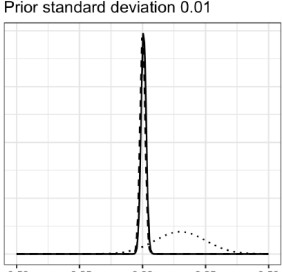
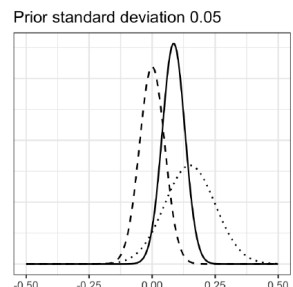
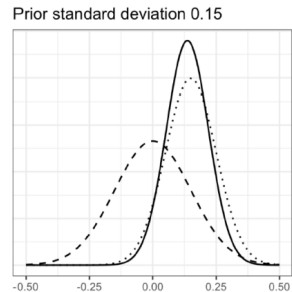

**Figure 2.** Visualization shrinkage prior based on a normal prior distribution with varying prior standard deviations. The dashed line indicates the prior, which is centered around zero. The dotted line indicates the likelihood, which is centered around 0.15. The solid line indicates the posterior, which is a compromise between the prior and the likelihood of the data. The smaller the prior standard deviation, the higher its peak around zero and, as a result, the posterior is pulled more towards zero.

### 3.1. Differences between Classical and Bayesian Regularized SEM

Aside from the general differences between the classical and Bayesian framework, for example, in terms of the interpretation of confidence versus credible intervals and the possibility to include prior information in the Bayesian framework, there are several differences especially relevant in the context of regularization. First, whereas classical estimation relies on optimization algorithms, the Bayesian framework generally uses Markov Chain Monte Carlo (MCMC) sampling to obtain the posterior distribution. The advantage of MCMC is that it results in a full posterior distribution, which also offers automatic uncertainty estimates for functions of parameters (such as indirect effects). The disadvantage of MCMC is that it can be slow, especially in high-dimensional models for which regularized SEM is particularly useful. Alternative Bayesian estimation algorithms exist that rely on approximations of the posterior distribution and are therefore computationally more efficient. I compare such an approximation to MCMC in the empirical illustration in Section 5.

Second, whereas classical regularization generally relies on cross-validation to determine the optimal value for the penalty parameter $\lambda$, the Bayesian framework offers multiple possibilities for determination of the parameter(s) that determine the amount of shrinkage (a notable exception is the work of [34] which proposes an automatic tuning procedure that does not rely on cross-validation). Note that shrinkage in the Bayesian framework will be influenced by one or more parameters of the prior distribution, depending on the specific shrinkage prior. For example, in the case of the normal shrinkage prior illustrated in Figure 2, there is only the prior variance that should be decided and that influences the shrinkage behavior. There are four ways to do so: (1) fix the prior variance to a specific value a priori; (2) determine the optimal value using cross-validation and consequently fix this value in the prior; (3) use an empirical Bayes estimate that estimates the value from the data and then fix this value in the prior (see, e.g., ref. [35]); or (4) estimate this value simultaneously with the model in a full Bayesian approach. Reference [33] provides a comparison of the empirical and full Bayesian approaches in the context of linear regression models. For the remainder of this article, the focus lies on options (1) and (4). The main advantage of the full Bayesian approach is that the specification of a hyperprior can result in a more robust marginal prior for the regularized effects. For example, when an inverse Gamma distribution is specified for the variance of the normal shrinkage prior, the resulting marginal prior distribution (after integrating out the variance) will be a Student's *t*-distribution. Compared to the normal distribution, Student's *t*-distribution has heavier tails and exerts less shrinkage on substantial effects.

### 3.2. Review of Developments in Bayesian Regularized SEM

The advantages of Bayesian regularized SEM have led various researchers to use Bayesian regularization in a specific SEM. Below, I discuss these applications of Bayesian

regularized SEM for different model types. An overview is provided in Table 1. As can be seen in Table 1, most work has focused on ridge or lasso types of priors, with the lasso extensions typically referring to adaptive or group lasso priors. A comprehensive overview of these priors and embedding within the general literature on shrinkage priors can be found in [33]. For a more practical tutorial on various shrinkage priors, the interested reader is referred to [36].

**Table 1.** Overview of the literature on Bayesian regularized SEM for different model types and shrinkage priors.

| | Ridge | Lasso | Lasso Extensions | Spike-and-Slab | Other |
|---|---|---|---|---|---|
| Exploratory factor model | | | | **Carvalho et al. (2008) [37]**; **Chen (2021) [38]**; West (2003) [39] | Bhattacharya & Dunson (2011) [40]; Conti et al. (2014) [41]; Legramanti et al. (2020) [42] |
| Confirmatory factor model | **Liang (2020) [43]** *; Lu et al. (2016) [44]; **Muthén and Asparouhov (2012) [45]** *; Vamvourellis et al. (2021) [46] | **Chen, Guo et al. (2021) [47]**; Pan et al. (2017) [48] | **Chen, Guo et al. (2021) [47]** | Lu et al. (2016) [44] | |
| Neural drift diffusion model | | Kang et al. (2022) [49] | | | |
| Item response model | Vamvourellis et al. (2021) [46] | **Chen (2020) [50]** | | | |
| Multiple-group factor model | **Shi et al. (2017) [51]** * | | Chen, Bauer et al. (2021) [52] | Chen, Bauer et al. (2021) [52] | |
| Non-linear SEM | | Guo et al. (2012) [53] | Brandt et al. (2018) [54]; Feng, Wang et al. (2015) [55] | Brandt et al. (2018) [54] | |
| General SEM | | Feng, Wu et al. (2015) [56]; Feng, Wu et al. (2017) [57] | Feng, Wu et al. (2015) [56]; Feng, Wu et al. (2017) [57] | | |
| Quantile SEM | | Feng, Wang et al. (2017) [58] | Feng, Wang et al. (2017) [58] | | |
| Latent growth curve model | | | Jacobucci and Grimm (2018) [26] | | |

Boldfaced references include user-friendly software implementations. * The methods used in these papers are available in `Mplus`.

### 3.2.1. Exploratory Factor Analysis

Refs. [37,39] considered a spike-and-slab prior to obtain sparse factor models with the main goal of dimension reduction in high-dimensional settings, specifically in gene expression studies. Based on [37], the software program `BFRM` was developed to run these types of models. In a similar setting, ref. [40] proposed a multiplicative gamma process prior, while [42] proposed a prior based on a sequence of spike-and-slab priors. Both are increasing shrinkage priors, meaning that they induce more shrinkage as the dimension grows, i.e., as the number of factors increases, the loadings are increasingly shrunken towards zero. Ref. [41] focused on a slightly lower-dimensional setting and explicitly incorporated identification criteria into their prior distribution, which is based on a hierarchical prior with Beta and Dirichlet hyperpriors. Ref. [38] also focused on a slightly lower-dimensional setting and proposed a spike-and-slab prior, which is implemented in the R-package `LAWBL`.

### 3.2.2. Confirmatory Factor Analysis

Bayesian regularized SEM has gained popularity among applied researchers through the work of [45]. They proposed the use of small-variance normal priors for the estimation of all cross-loadings or residual covariances simultaneously within a confirmatory factor model (note that with shrinkage priors, confirmatory factor analysis is no longer strictly confirmatory since we do not assume the simple structure as in traditional CFA. However, I still use the term CFA to distinguish from a fully exploratory approach in which no main loadings are specified). Although not the primary target of their article, they also illustrated the use of small-variance normal priors for both structural and measurement parameters in an SEM. Although they did not connect this approach to ridge regularization, the normal prior is known to be equivalent to the ridge penalty [28]. Importantly, through implementation of this approach in `Mplus`, their work has enabled applied researchers to easily use the methodology. Ref. [44] placed the work by [45] in the regularization framework by illustrating the connection between the small-variance normal prior and ridge regularization. They expanded on this work by considering a spike-and-slab prior as well and compared both approaches, illustrating how different prior distributions lead to slightly different factor solutions. Ref. [43] investigated the influence of the prior variance choice for the small-variance normal priors from [45] on cross-loadings on model fit, population model recovery, true and false positive rates, and parameter estimation. Ref. [46] extended the proposed approach by [45] to handle alternative distributions, including for example, the logistic distribution to accommodate item response models, and proposed an approach based on out-of-sample predictive performance to assess goodness of fit. Another application to item response models was provided by [50] based on the Bayesian lasso. In a continuous confirmatory factor analysis setting, ref. [47] presented a lasso and covariance lasso approach to (potentially simultaneously) detect relevant cross-loadings and residual covariances. This is related to the work by [48] who proposed a covariance lasso prior for the inverse of the covariance matrix to automatically obtain a sparse, yet positive, definite residual covariance matrix. Ref. [59] extended the work of [47] by allowing for mixed data types and missing data. In addition to the lasso for cross-loadings and the covariance lasso for residual covariances, ref. [59] proposed the adaptive lasso. All methods are implemented in the R-package `LAWBL`.

### 3.2.3. Non-Linear SEMs

Ref. [53] proposed a Bayesian lasso to model non-linear relations among latent variables in semiparametric SEMs. Ref. [55] improved upon the work by [53] by considering an adaptive group lasso prior that can handle group effects introduced by the basis expansions and adaptively penalize different groups of coefficients. Ref. [54] combined the adaptive lasso with a spike-and-slab prior to obtain a more flexible approach to model non-linear effects that outperformed a standard Bayesian lasso, especially in situations with high multicollinearity or low reliability.

### 3.2.4. Other Models

Ref. [49] investigated the Bayesian lasso in the context of a factor analysis neural drift diffusion model that is used to simultaneously analyze behavioral and neural data, resulting in high-dimensional data for which regularization is especially useful. Ref. [51] used the small-variance normal priors from [45] to identify non-invariant parameters in the context of multiple-group factor analysis. They first focused on identifying a proper reference indicator by considering which item has the highest likelihood to be invariant across groups. Ref. [52] proposed and discussed the use of spike-and-slab priors to detect measurement non-invariance, which offers the advantage of more theoretically coherent parameter selection based on posterior inclusion probabilities. Their specific implementation combined the spike-and-slab prior with the double-exponential or lasso prior similar to [54]. Ref. [56] presented a lasso and adaptive lasso implementation for ordinal regression with latent variables (where the latent variables are measured using continuous indicators).

Ref. [57] implemented a Bayesian lasso and adaptive lasso in a generalized latent variable model that can accommodate mixed data types. Note that this work is different from [59] in that [57] considered regularization on the regression coefficients with the latent variables as potential regressors, whereas [59] considered regularization in the measurement model (i.e., on the cross-loadings or residual covariances). Ref. [58] proposed Bayesian lasso and adaptive lasso priors for quantile regression coefficients in the structural part of SEMs. In their comparison of classical and Bayesian regularized SEM, ref. [26] considered a latent growth curve model in which the Bayesian adaptive lasso was used to shrink and identify relevant covariates. The results from the Bayesian adaptive lasso were compared to those from the classical lasso.

## 4. An Overview of Software Packages for Regularized SEM

I next present an overview of available software packages for regularized SEM and their capabilities. Although the focus lies specifically on user-friendly packages for Bayesian regularized SEM, I also discuss three popular user-friendly R-packages for classical regularized SEM as well as general software packages that enable the user to specify regularized structural equation models, but do not provide out-of-the-box implementations of these models. An overview of all software packages is available in Table 2. Packages in italics are included in the comparison in Section 5.

**Table 2.** Overview of software packages available for regularized SEM.

| Package | Open-Source | Free | User-Friendly [1] | Model Flexibility | Penalty/Prior Flexibility |
|---|---|---|---|---|---|
| Classical regularized SEM | | | | | |
| regsem | + | + | + | + | + |
| *lslx* | + | + | + | + | + |
| penfa | + | + | + | − | + |
| lessSEM | + | + | ∼ | + | + |
| Bayesian regularized SEM | | | | | |
| Mplus | − | − | − | + | − |
| OpenBUGS/JAGS | + | + | − | + | + |
| *Stan* | + | + | − | + | + |
| PyMC3 | + | + | − | + | + |
| Greta | + | + | −/+ | + | + |
| TensorSEM | + | + | − | + | + |
| *blavaan* | + | + | + | + | − |
| *LAWBL* | + | + | + | − | − |
| blcfa [2] | ∼ | ∼ | ∼ | − | − |
| infinitefactor | + | + | + | − | + |

[1] "User-friendly" refers specifically to whether applied researchers with some knowledge of R are able to use the software, i.e., whether models are implemented without the user needing to learn special syntax apart from R and potentially `lavaan` syntax. [2] Although the R-package `blcfa` itself is free, open-source, and easy to use, it interacts with `Mplus`, which does not have these properties.

### 4.1. General Purpose Software That Is Able to Perform Bayesian Regularized SEM

Arguably the most popular software program for SEM is `Mplus` [60] (although other software programs for SEM such as `Amos` have implemented Bayesian estimation, these packages rely on default, flat prior distributions and do not allow the user to specify shrinkage priors for regularization). `Mplus` is a closed-source, paid software program that allows the specification of a wide variety of SEMs, ranging from simple path models, to (multilevel) factor models, as well as dynamic SEM. Various classical estimation algorithms and Bayesian MCMC sampling are implemented, and it is through the latter that Bayesian regularized SEM is possible in `Mplus`. One major drawback of `Mplus` is that it is limited in terms of the types of prior distributions that can be considered. Specifically, in the context of regularized SEM, only conjugate normal priors are available for location parameters such as loadings or regression coefficients, and thus, only ridge-like shrinkage behavior can be achieved. Moreover, it is not possible to estimate the variance of the normal prior within

the model so instead the value of this hyperparameter needs to be fixed to a specific value by the researcher (generally a small value such as $\sigma_0^2 = 0.01$ or $\sigma_0^2 = 0.001$). As a result, all regularized parameters will be shrunken heavily by the same amount.

There exist various Bayesian software packages that allow the user to specify a probabilistic model, which is then estimated using some form of Markov Chain Monte Carlo (MCMC) sampling. The oldest package is `BUGS` (Bayesian Inference using Gibbs Sampling; [61]), with `WinBUGS` [62] being the Microsoft Windows version. Currently, `WinBUGS` is no longer being actively developed, but the open-source version `OpenBUGS` [63] is.

`OpenBUGS` can be run from Microsoft Windows or Linux, or through R. A similar package, `JAGS` (Just Another Gibbs Sampler; [64]), uses a very similar programming language but is platform independent. As the names imply, the `BUGS`/`JAGS` family of software packages relies on Gibbs sampling [65] to obtain the posterior distribution.

`Stan` [66] is a more recent software package for general probabilistic programming that relies on the No-U-Turn sampler (NUTS; [67]), a variant of Hamiltonian Monte Carlo (HMC; see e.g., ref. [68]) instead of Gibbs sampling. For an extensive comparison between `BUGS` and `Stan`, see the Stan user's guide [66]. For our purpose, it suffices to say that in general, HMC is more efficient compared to Gibbs sampling, especially for complex models. In addition, it provides a wider variety of convergence diagnostics. One drawback of `Stan`, however, is that it currently does not allow the use of discrete parameters, making the implementation of, for example, a spike-and-slab prior, difficult. Similar to `Stan`, `PyMC3` [69] also relies on NUTS but is specifically for Python, while `Stan` interfaces with multiple languages, including Python.

Both `Stan` and `PyMC3` have approximate algorithms implemented based on variational inference to more efficiently approximate the posterior distribution. Whereas MCMC sampling draws directly from the posterior distribution, variational inference approximates the posterior using a simpler distribution. It does so by searching over a family of simple distributions and subsequently finding the distribution that is closest to the posterior according to some criterion. The advantage over MCMC sampling is that variational inference is typically faster and able to scale better to high-dimensional data sets. The disadvantage is that the approximation might be far from the true posterior distribution.

Two R-packages that rely on machine learning libraries are `greta` [70] and `TensorSEM` [71]. `Greta` relies on `TensorFlow`, while `TensorSEM` relies on `Torch`. `Greta` is designed for Bayesian modeling and offers a lot of flexibility in terms of models and priors that can be considered. `TensorSEM` is more general and considers SEM as a computation graph that enables a flexible way to extend traditional SEMs, for example, through the specification of a penalty or shrinkage prior. Both packages require the user to specify the model in a specific way, either through probabilistic statements directly in R (in the case of `Greta`) or using `lavaan` syntax. In addition, `TensorSEM` requires the user to be familiar with `Torch`. These packages, as well as `Stan` and the `BUGS`/`JAGS` programs, are extremely helpful in advancing theoretical developments in Bayesian regularized SEM due to their flexibility, but they are not user-friendly enough to be used generally by applied researchers.

*4.2. Software Packages for Classical Regularized SEM*

Ref. [3] implemented a classical approach to regularized SEM utilizing Reticular Action Model (RAM) notation in the R-package `regsem` [5]. The syntax in the `regsem` package relies on `lavaan` [72] and the models supported in `regsem` are therefore limited by the models available in `lavaan`. Current exceptions are multiple-group models and models with categorical variables that are not supported in `regsem`. In terms of penalty functions, `regsem` has implemented a wide variety including the ridge [28], lasso [27], and elastic net [29], and generalizations of the lasso such as the adaptive lasso [73]. Furthermore, the smooth clipped absolute deviation (SCAD; [74]) and minimax concave penalty (MCP; [75]) can be used to obtain results that are sparser compared to the lasso. Ref. [4] developed a penalized likelihood method for SEM and implemented this in the R-package `lsl`, which is the predecessor of the more extensive and improved package `lslx` [6]. `lslx`

also relies on `lavaan` syntax for model specification but improves upon `regsem` in terms of convergence rates and speed. Note that in certain cases, for example, when there is high multicollinearity present, `lslx` can also suffer from convergence issues [76]. `lslx` allows the user to specify either a ridge, lasso, elastic net, or minimax concave penalty (MCP). Whereas `regsem` relies on general non-linear optimization algorithms or coordinate descent, `lslx` employs a modified quasi-Newton algorithm. Finally, ref. [34] implemented a trust-region algorithm in which they locally approximated non-differentiable penalties including the lasso, adaptive lasso, SCAD, and MCP. This algorithm has been implemented in the R-package `penfa`. The advantage of their implementation is that it offers automatic tuning of the penalty parameters, and as a result, the user does not need to manually specify values for cross-validation. `penfa` only supports single- and multiple-group factor analysis. Recently, the R-package `lessSEM` has been published [77]. This package has been heavily influenced by `regsem` and `lslx`; however, its focus is more on method developers to aid them in the development of new regularized SEMs rather than applied researchers using regularized SEM. Therefore, the user-friendliness is denoted as in Table 2.

*4.3. User-Friendly Software Packages for Bayesian Regularized SEM*

There exist three user-friendly R-packages for Bayesian regularized SEM. First, `blavaan` [78] offers Bayesian estimation of SEMs with syntax and functions similar to `lavaan`. It can therefore handle the same types of models as `lavaan`, with the exception of multilevel SEMs. The original versions of `blavaan` first generated `JAGS` and later also `Stan` code at runtime based on the specified model. As of Version 0.3, `blavaan` relies on a precompiled `Stan` model based on the marginal likelihood. Although computationally more efficient, this approach is restricted in that distributional forms of priors are generally fixed (only hyperparameters can be changed) and there are certain restrictions with respect to equality constraints. For the application of Bayesian regularized SEM, the first restriction implies that only normal ridge priors with a fixed standard deviation can be used in `blavaan`. A workaround would be to utilize the previous, slower MCMC implementations or to export the `Stan` file underlying `blavaan` and directly edit the priors. Second, ref. [59] presented a general partially confirmatory approach to SEM relying on the (adaptive) lasso for shrinkage of cross-loadings and the graphical or covariance lasso [79,80] for relaxation of the local independence assumption (i.e., through shrinkage of residual covariances). An MCMC sampling algorithm is implemented in the R-package `LAWBL` (Latent (variable) Analysis With Bayesian Learning; [81]) that currently supports factor models and item response models. Note that the Bayesian covariance lasso is also implemented in the R-package `blcfa` [82] which allows detection of significant residual covariances and subsequently frees these covariance parameters and estimates the subsequent model using `Mplus`. Thus, this package requires `Mplus` to be installed. Third, the `infinitefactor` package [83] implements the multiplicative Gamma shrinkage prior proposed by [40] as well as Dirichlet–Laplace priors [84]. This package is specifically designed for exploratory factor analysis in high-dimensional settings.

## 5. Empirical Illustration: Regularizing Cross-Loadings in Factor Analysis

The goal of this empirical example is twofold: (1) to illustrate similarities and differences between implementations of Bayesian regularized SEM in existing software packages; and (2) to aid applied researchers in using these software packages and applying Bayesian regularized SEM in their own work. All code to run this example is available at the author's github (https://github.com/sara-vanerp/BRSsoftware). In addition, there is a Markdown file (https://github.com/sara-vanerp/BRSsoftware/tree/main/appendix) that summarizes the main functions needed to use each package (accessed on 1 July 2023).

To obtain a realistic data set that can be freely shared online, I generated a data set based on the observed covariance matrix from a study on the adaptive ability performance test (ADAPT; [85]). The ADAPT consists of 65 items intended to assess adaptive skills in individuals with intellectual disabilities and borderline intellectual functioning across

three domains (conceptual, social, and practical). A data set with $N = 1000$ observations was generated by drawing from a multivariate normal distribution with means zero and covariance matrix equal to the complete observed covariance matrix and subsequently splitting into a training ($N = 748$) and test ($N = 252$) set. For illustrative purposes, I estimated a model with three factors (with 36 items) based on the full solution. This resulted in some items being included with a main loading of only 0.30, which is not ideal but does provide us with the type of realistic data one may encounter in practice. A preliminary analysis including all items resulted in convergence problems for `regsem` and long run times for `blavaan` (around 45 min per analysis). This illustrates a constraint of the current implementations of regularized SEM in that, despite these methods being especially useful in larger models, these larger models often have difficulties converging without careful tuning of cross-validation parameters or prior hyperparameters, which is impractical given the long run times. The model is shown in Figure 3.

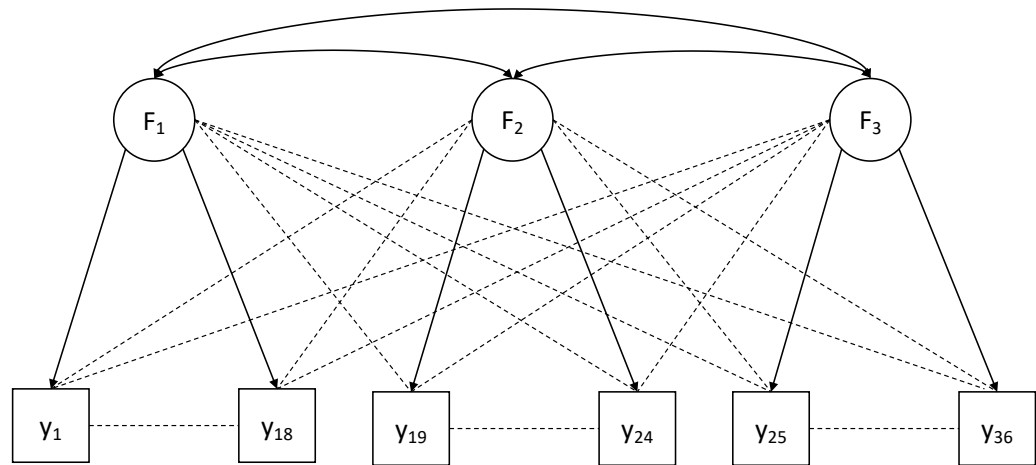

**Figure 3.** The model used in the empirical application. Dashed lines between indicators reflect that not all variables are visualized, but the subscripts indicate the number of items assumed to have a main loading on each factor. All cross-loadings are estimated with regularization, as reflected by the dashed arrows. In total, 72 cross-loadings were estimated in addition to the 36 main loadings.

I focus primarily on user-friendly and open-source software packages for Bayesian regularized SEM, although results from `lslx` using the lasso, elastic net, and minimax concave penalties are included for comparison. Of the available Bayesian software packages, only `blavaan` and `LAWBL` are user-friendly, open-source options for regularization of cross-loadings (`infinitefactor` focuses on completely exploratory settings in which all loadings are regularized; here the aim is only to regularize the cross-loadings) so these two packages are compared. For `blavaan`, prior variances equal to $\sigma_0^2 = 0.1$, $\sigma_0^2 = 0.01$, and $\sigma_0^2 = 0.001$ were considered. Note that the normal prior in `blavaan` is parametrized such that the standard deviation $\sigma_0$ should be provided rather than the prior variance $\sigma_0^2$. In addition, both the MCMC algorithm and the variational Bayes algorithm were compared. For `LAWBL`, the lasso and adaptive lasso were compared. Finally, three more elaborate prior distributions were included in the comparison: the ridge prior with the standard deviation estimated instead of fixed, the regularized horseshoe prior with a default prior setting, and the regularized horseshoe with a prior guess for the number of substantial cross-loadings (based on estimated cross-loadings $\lambda > 0.20$ in the initial EFA). These priors are not available in user-friendly software packages so they have been coded manually in Stan. Code for these models is available online and can be adapted for other applications.

*5.1. Before the Analysis*

In any Bayesian analysis, it is considered good practice to understand the prior distribution that you are using. In Bayesian regularized SEM, visualizing the shrinkage prior

can provide the researcher with an idea of the shrinkage behavior that can be expected. Figure 4 shows the densities of the prior distributions compared in the illustration. The horseshoe prior goes to infinity at zero and will therefore heavily shrink small effects. However, its heavy tails allow substantial effects to escape this shrinkage. The lasso, on the other hand, has thinner tails and, as a result, will exert more shrinkage towards zero on substantial effects. One way to avoid this is by using an adaptive lasso in which each parameter has a local shrinkage parameter in addition to the global shrinkage parameter that affects all cross-loadings. These parameter-specific local shrinkage parameters allow more adaptive shrinkage behavior in which substantial effects are shrunken less towards zero compared to small effects. For the ridge prior, the heaviness of the tails depends on whether the standard deviation is estimated (resulting in heavy tails) or fixed to a specific value (resulting in thin tails). The latter option, which is available in `blavaan`, results in theoretically suboptimal shrinkage behavior because it shrinks all effects heavily towards zero, including substantial ones.

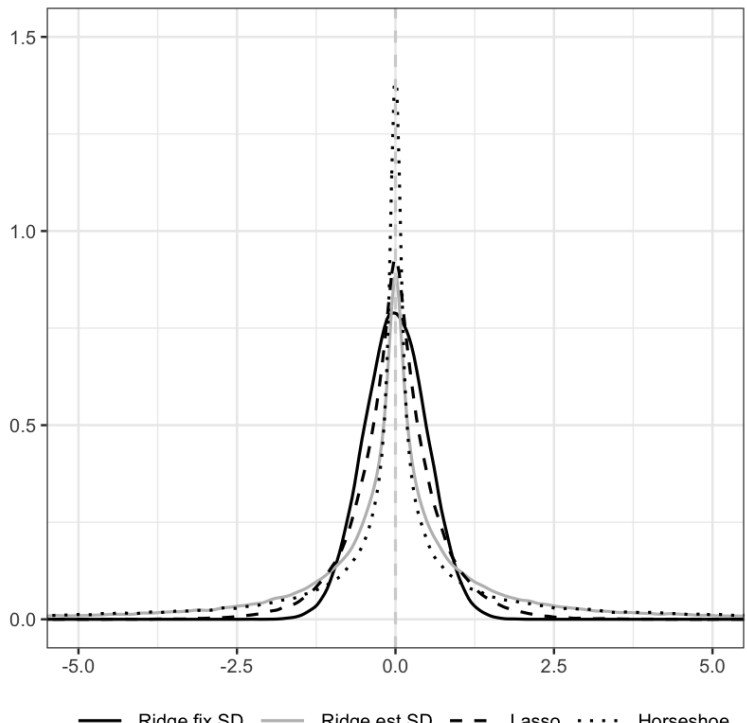

**Figure 4.** Comparison of the shrinkage priors in the illustration. The ridge corresponds to a normal prior in which the standard deviation is either estimated ("est SD"; grey line) or fixed to some value ("fix SD"; black line). The vertical dashed grey line at zero reflects the asymptote of the horseshoe prior.

In addition to the shrinkage priors on the cross-loadings, all other parameters in the model should be given a prior distribution as well. Software packages generally implement default prior distributions that are meant to be non-informative. Although in most situations these default priors are indeed non-informative, this might not always be the case [86], and so it is good to be aware that prior distributions are specified for the nuisance parameters and which distributions are being used. In `blavaan`, the default priors can be requested through the function `dpriors()` and it is possible to change the hyperparameters for these default priors, for example, to incorporate prior knowledge for certain parameters. For `LAWBL`, only the distributional forms for the prior distributions are reported in [59], which are conjugate priors. However, the exact values for the hyperparameters are unclear and it is not possible to change these values.

Note that regularization allows the estimation of parameters that would not be identified in traditional SEM. In this application, we estimate all cross-loadings in addition to the main loadings and factor correlations. Including a penalty function or shrinkage prior that pulls small estimates sufficiently towards zero ensures identification of the model. However, we still need to impose additional identification constraints to identify the latent variables by either setting one (main) loading for each latent variable to 1 (unit loading identification) or each latent variable variance to 1 (unit variance identification). By default, `lslx` and `blavaan` use unit loading identification by fixing the first main loading of each factor, while `LAWBL` uses unit variance identification.

An important step when applying regularized SEM is to scale the data before analysis. In particular, shrinkage priors with only a global shrinkage parameter such as the ridge and lasso will penalize variables differently when they are measured on different scales. This is a general feature of prior distributions: the amount of influence they exert on the results depends on their scale in relation to the scale of the data. I therefore scaled the data such that each item had a mean of zero and a variance of one before applying the (Bayesian) regularization methods.

*5.2. During the Analysis*

Within the Bayesian framework, MCMC sampling is generally used to obtain the posterior distribution on which inference is based. Although there exist many different MCMC algorithms (e.g., Gibbs, Metropolis-Hastings, Hamiltonian Monte Carlo), they all share the property that they draw samples from the posterior distribution. Enough posterior draws need to be sampled to obtain a reliable representation of the posterior distribution. Depending on the correlation between subsequent draws, this can take quite some time. However, it differs between algorithms, and thus between software packages, how efficient a sampler is, and thus how many draws are needed. Generally, the HMC algorithm implemented in `Stan` is more efficient compared to Gibbs samplers in that the posterior draws exhibit less autocorrelation. In addition, convergence needs to be assessed to ensure that the sampler has converged to a stationary distribution. There are different ways of assessing convergence and different software packages have different convergence diagnostics implemented, see, for example [87], for a tutorial on different ways of assessing convergence. Some general guidelines when assessing convergence are:

- Run multiple chains starting from different starting values. Traceplots can be used to visually assess whether the different chains coincide at some point and mix well (i.e., the traceplots should resemble "fat caterpillars").
- Remove a specified number of initial iterations ("burn-in") to avoid the final results depending on the starting values. Traceplots based on the iterations without burn-in should immediately indicate nice mixing.
- Consider numerical convergence diagnostics such as the potential scale reduction factor [88], which should be close to 1 (`EPSR` in `LAWBL` and `Rhat` in `blavaan`).
- Make sure you have a sufficient number of effective samples (`Neff` in `blavaan`). Although there are no theoretically derived guidelines, a useful heuristic that has been proposed would be to worry whenever the ratio of effective sample size to full sample size drops below 0.1 [89].
- When in doubt, or when not all diagnostics are (easily) available, you can rerun the analysis with double the number of iterations to ensure stability of the results.

However, the extent to which these guidelines can be (easily) followed depends on the packages used. Of the packages compared in this illustration, `blavaan` offers many convergence diagnostics as well as warnings when convergence appears problematic. `LAWBL` requires the user to manually assess convergence, for example, by running multiple MCMC chains from random starting values, creating traceplots, and computing convergence diagnostics. Please see the code and Markdown file available online for functions to assess convergence.

Given the computational cost of MCMC, alternative algorithms have been developed to obtain approximations of the posterior distribution. Specifically, `Stan` (and thus `blavaan`) has the Automatic Differentiation Variational Inference (ADVI; [90]) algorithm implemented, which uses a flexible approximating distribution and subsequently minimizes the difference between this approximation and the true distribution. An important quantity in this algorithm is the ELBO (Evidence Lower Bound), which is optimized to find the best possible approximation to the posterior distribution using Monte Carlo integration. Thus, variational Bayes in Stan also requires the user to specify a number of draws (`elbo_samples`), but this is not the number of draws sampled from the posterior but rather the number of draws used to approximate the ELBO. See Stan's User Manual [66] for more information on this algorithm. It is important to note that the variational Bayes algorithm in `Stan` is still considered an experimental feature and thus not recommended for final inference. The ADVI algorithm was included in the comparison to assess the quality of this approximation for the empirical application.

### 5.3. After the Analysis

In this section, I discuss a selection of results from the empirical example to illustrate various features of the Bayesian regularized SEM methods that were compared. All results are available online (https://github.com/sara-vanerp/BRSsoftware, (accessed on 1 July 2023)).

#### 5.3.1. Comparison Variational Bayes and MCMC

A comparison of the variational Bayes algorithm versus the MCMC NUTS algorithm in `Stan` illustrates the experimental character of this feature. For the normal ridge priors with fixed standard deviations as implemented in `blavaan`, in particular, the setting where the prior variance equals 0.1 led to much larger estimates overall for the cross-loadings compared to the MCMC algorithm and the other prior specifications. This difference became even more pronounced for the manual ridge implementation in `Stan` where the standard deviation is estimated. For this prior, all cross-loadings were extremely large with many being estimated to be around 3. The regularized horseshoe also showed differences between the two algorithms, with the extent of the differences depending on the exact prior specification. See Figure 5 for an example.

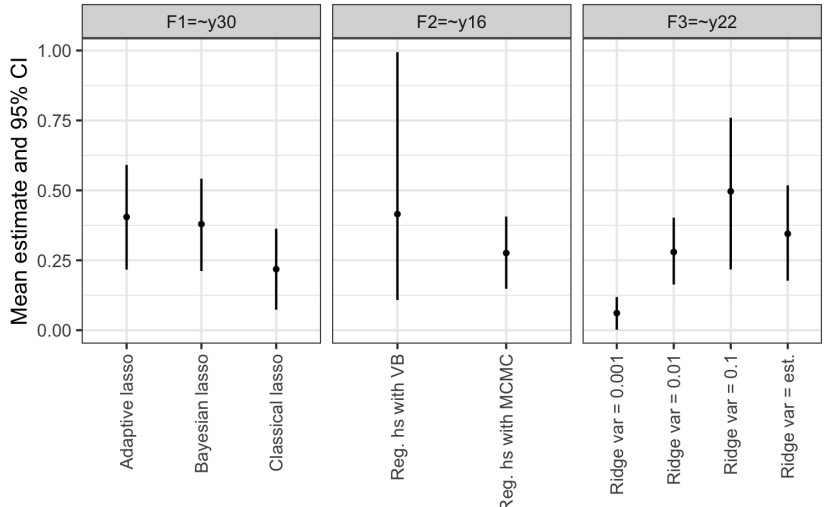

**Figure 5.** Estimates and 95% confidence or credible intervals for selected shrinkage methods and selected cross-loadings. "Reg. hs" stands for the regularized horseshoe prior, "VB" stands for the variational Bayes algorithm, and "MCMC" stands for Markov Chain Monte Carlo. The ridge corresponds to a normal prior with its variance fixed to a specific small value or estimated ("var = est.").

5.3.2. Posterior Estimates Cross-Loadings

Overall, plots of the posterior estimates and credible intervals show immediately which cross-loadings are probably substantial and which cross-loadings seem irrelevant. For example, in Figure 6, it is clear that cross-loadings $F_1 =\sim y_{28}$ and $F_1 =\sim y_{30}$ were estimated to be substantial, whereas loadings $F_1 =\sim y_{29}$, $F_1 =\sim y_{31}$, and $F_1 =\sim y_{32}$ were shrunken completely towards zero and thus can be seen as irrelevant. However, for cross-loadings $F_1 =\sim y_{27}$ and $F_1 =\sim y_{33}$, it is unclear. The posterior mean estimates were slightly away from zero, but the 95% credible intervals did contain zero. If we wish to decide which cross-loadings can be put to zero, we need some cutoff to do so based on these results. Different cutoffs have been proposed, for example, using a threshold for the point estimate, or the 95% credible interval, but as shown by [91], it depends on the condition as to which specific threshold or cutoff performs best, making it difficult to apply these cutoffs in practice. Classical regularization approaches do have the potential to estimate parameters to be exactly zero, thereby forgoing the need for arbitrary cutoffs. However, this leaves the issue of post-selection inference. Specifically, after selecting relevant parameters through regularization, subsequent inference is conditional on the model selection, which affects sampling distributions of the resulting parameter estimates. As a result, traditional naive inference methods can be invalid and more advanced alternatives are needed (see e.g., [92,93]).

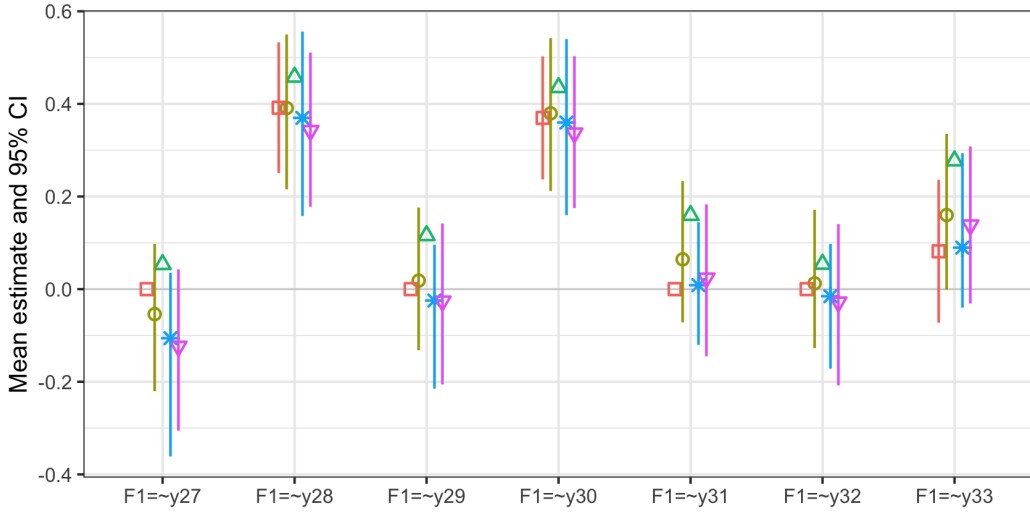

**Figure 6.** Estimates and 95% confidence or credible intervals for selected shrinkage methods and selected cross-loadings. "EFA" is traditional exploratory factor analysis. "Reg. hs MCMC" stands for the regularized horseshoe prior using Markov Chain Monte Carlo sampling. "Ridge var = est." stands for the normal prior with the variance estimated.

As we would expect based on the different distributional forms of the priors, the resulting shrinkage behavior on the cross-loading estimates varied. This is seen most clearly for the normal ridge priors with fixed variances (see Figure 5). The results shown in Figure 5 for selected cross-loadings are exemplary for all cross-loadings: as the prior variance decreases and the prior becomes more peaked around zero, it exerts more influence, thus shrinking the estimates more heavily towards zero. The estimates based on the ridge prior with estimated variance tended to lie between the estimates from the ridge prior with variances equal to 0.01 and 0.1. This is not surprising given that the posterior mean of the estimated variance parameter for the ridge prior was equal to 0.028. Note that this estimate is informed by the data, which is an advantage of this implementation. Interestingly, the posterior mean estimates did not differ substantially between the lasso and adaptive

lasso implementations in `LAWBL`, although the adaptive lasso resulted in slightly larger 95% credible intervals (defined as the highest density interval; see Figure 5). Notably, for the regularized horseshoe, results were exactly the same regardless of whether the default specification was used (global scale = 1) or whether a prior guess for the number of substantial cross-loadings was provided (leading to a global scale of 0.007).

Similarly, different classical penalty functions led to different shrinkage behaviors. Generally, if a cross-loading was estimated to be zero, this conclusion was reached across the lasso, elastic net, and minimax concave penalties. However, relevant cross-loadings tended to be estimated at a larger value for the minimax concave penalty compared to the lasso and elastic net. Thus, similar to the horseshoe and other heavy-tailed shrinkage prior distributions, the minimax concave penalty seems to allow substantial effects to be larger compared to other penalties that correspond to thinner-tailed shrinkage priors. Finally, it is worth noting that the classical lasso and the Bayesian lasso did not necessarily lead to similar results and can even lead to quite different estimates, as shown in Figure 5. This is because, even though the double-exponential or Laplace prior distribution can be shown theoretically to correspond to a lasso penalty, this equivalence only holds in practice when the value of the hyperparameter in the shrinkage prior equals the value of the penalty parameter in the penalty function and when the posterior mode is used [94].

### 5.3.3. Nuisance Parameters

Although the main interest in this application lies in the estimation and selection of relevant cross-loadings, it is important to note that due to the interconnectedness of the parameters in SEM, the shrinkage priors will affect estimates for the other parameters in the model as well. This is shown in Figure 7 for the factor correlations, with differences between estimated correlations being as large as 0.43 between certain priors. Note that these differences can arise indirectly, from differences in estimated cross-loadings due to the shrinkage priors, as well as directly, from differences in priors on the factor correlations themselves. In this application, default priors were used for the nuisance parameters, which are meant to be non-informative.

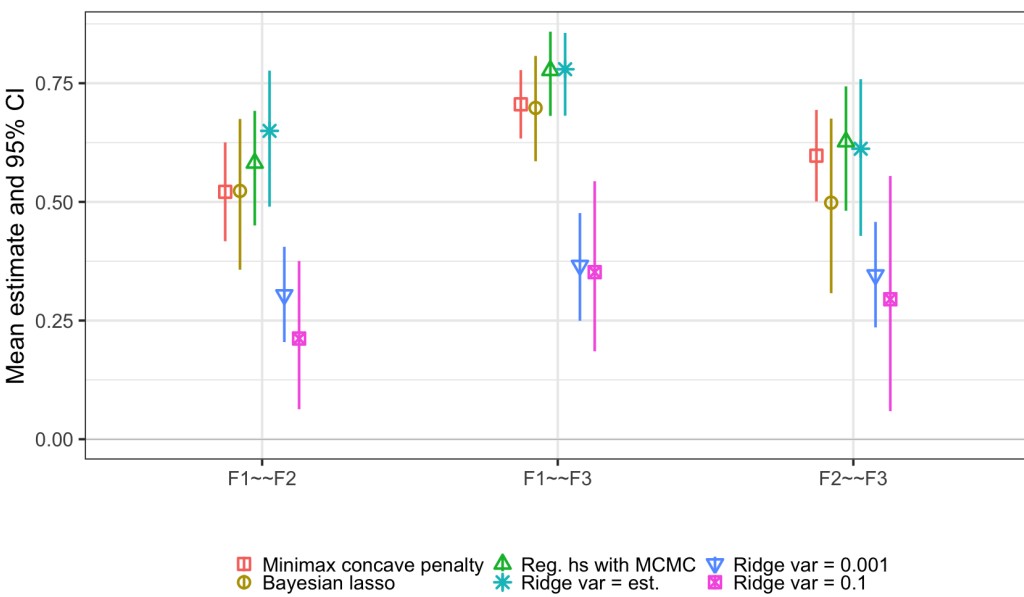

**Figure 7.** Estimates and 95% confidence or credible intervals for selected shrinkage methods and factor correlations. "Reg. hs MCMC" stands for the regularized horseshoe prior using Markov Chain Monte Carlo sampling. "Ridge var = est." stands for the normal prior with the variance estimated.

## 6. Discussion

This paper presented an overview of the current state of research on Bayesian regularized SEM. Compared to the traditional divide between exploratory and confirmatory research, regularized SEM can be seen as bridging these two extremes (see also the comparisons by [95,96] of Bayesian SEM and exploratory SEM). Depending on the specific penalty or shrinkage prior used, it lies closer to one or the other end of the continuum. In general, researchers might consider using regularized SEM whenever their model has many parameters, of which they expect a subset to be equal to zero. Depending on the desired shrinkage behavior, different shrinkage priors can be used. For example, if a large number of substantial parameters is to be expected, it might be needed to use a shrinkage prior that offers non-negligible shrinkage to zero for relevant effects to avoid identification issues. The empirical application considered in this paper has been kept, for illustrative purposes, relatively simple. More advanced applications are possible, although software implementations might be limited.

The literature review indicated that theoretical work on Bayesian regularized SEM has focused mainly on ridge, lasso-type, and spike-and-slab priors. These priors have been proposed for a wide variety of different types of models, with most work focusing on factor analysis. However, a review of available software packages showed that the applicability of Bayesian regularized SEM remains limited by the available (user-friendly) software implementations, which are restricted to relatively simple models and shrinkage priors. An empirical example provided an illustration of using various (Bayesian) regularization methods to identify substantial cross-loadings in factor analysis to aid researchers in applying Bayesian regularized SEM in their own work and to highlight various similarities and differences in Bayesian shrinkage methods.

The empirical example illustrated some differences between the currently available simple shrinkage priors such as the ridge with fixed variance and lasso and more advanced options such as the regularized horseshoe and ridge with estimated variance. An overview of these and other popular shrinkage priors in the context of linear regression models showed differences in shrinkage behavior as well as predictive and selection performance across priors [33] and so differences can be expected in the context of SEM as well. Future work should investigate which priors perform best in which models and under which conditions. It is important to note that `blavaan` also provides fit indices and posterior predictive $p$-values to assess model fit. However, ref. [97] showed that the posterior predictive $p$-value is not suited for the evaluation of models based on small-variance priors as it performs inconsistently as the sample size increases. Ref. [98] developed Bayesian alternatives for commonly used fit indices that are reported by `blavaan`. However, their illustrative examples provided some preliminary support for informative priors influencing fit indices. Although more extensive simulations are needed in this area, it is to be expected that the shrinkage priors considered in this paper will also influence the fit indices, which raises the question as to what these indices assess: they are no longer an evaluation of the fit of strictly the model to the data, but they become an evaluation of the fit of the posterior, which includes the prior, to the data. An open question is therefore how researchers should evaluate the fit of Bayesian regularized SEMs, especially because shrinkage priors are by design introducing bias in the estimation procedure to obtain a more generalizable model. This aim of balancing the bias and variance that is inherent in regularized SEM should be taken into account when assessing the final model, for example, using model assessment approaches that monitor out-of-sample predictive performance such as the method proposed by [46].

### Future Directions

Given the findings presented here, I identify three important areas for future research. First, the ultimate selection of relevant parameters requires attention. Current methods, such as the use of thresholds for the posterior estimates or credible intervals lead to relatively arbitrary cutoffs with the optimal cutoff depending on the specific data-generating

conditions [91]. In addition, current methods rely on the marginal posterior distribution, which can behave differently compared to the joint posterior. Thus, future research should develop and investigate alternative methods of variable selection that jointly select parameters (such as projection predictive variable selection [99], or decoupled shrinkage and selection [100]).

Second, the development of user-friendly software that incorporates various different shrinkage priors and models would enable more flexible shrinkage behavior for a wider variety of SEMs. Ideally, such a software package builds upon existing packages, such as `blavaan`, so that users can rely on familiar syntax. One difficulty with such a general purpose software implementation is that the shrinkage priors should be implemented using reasonable default values for the hyperparameters. A difficulty is that the resulting prior specification should be restrictive enough such that the model is identified in a classical sense, but not so restrictive as to inadvertently exert too much influence on the results. Although it is always a good idea to assess the sensitivity of the results to the specific choice of the prior [86], reasonable default values will aid applied researchers in using Bayesian regularized SEM. In addition, guidelines should be developed for applied researchers to translate their prior beliefs into reasonable specifications of the shrinkage prior. Ref. [44] notes that "different prior choices represent different beliefs about the data at hand and no single prior distribution should be considered superior in all situations. Substantive researchers are advised to pick prior distributions that fit their prior beliefs concerning the data analytic problem at hand", but does not provide further guidelines on how to do so.

Third, the use of (Bayesian) regularized SEM is especially advantageous in high-dimensional models. Apart from some of the work on exploratory factor analysis, most research on Bayesian regularized SEM focuses on lower-dimensional settings. This focus is most likely due to the computational cost of MCMC, so to fully unlock the potential of Bayesian regularized SEM, scalable algorithms need to be available. The ADVI algorithm in `Stan` offers one such example; however, a preliminary investigation of this experimental algorithm in the empirical illustration showed that it can lead to results that differ greatly from the results obtained using MCMC. Future work should look into the quality of approximations of the posterior distributions obtained using variational Bayes algorithms in high-dimensional SEMs or develop more scalable full MCMC algorithms (for example, based on the spike-and-slab prior; [101]).

**Funding:** This research received no external funding.

**Institutional Review Board Statement:** Not applicable.

**Informed Consent Statement:** Not applicable.

**Data Availability Statement:** All data and code used in this manuscript are available from the author's Github: https://github.com/sara-vanerp/BRSsoftware (access on 1 July 2023).

**Acknowledgments:** The author would like to thank David Goretzko for providing useful references and discussions on traditional regularized SEM, including the different software packages available. The author would also like to thank Edgar Merkle for providing useful comments on a preprint of this manuscript and to the reviewers for their input.

**Conflicts of Interest:** The author declares no conflict of interest.

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
