# Peer review of "Bayesian Regularized SEM: Current Capabilities and Constraints"

_psych, doi:10.3390/psych5030054_

Round 1

Reviewer 1 Report

This study reviews the regularized SEM in both frequentist and Bayesian frameworks with the purpose to provide applied users with some training on Bayesian regularized SEM. The study is more like a tutorial paper on how to use regularized SEM. However, it does not provide the necessary details for fitting regularized SEM and for Bayesian inference.  I would suggest including more details for future revision.

1. The paper mentions the term "variational Bayes" without providing an explanation of how it works or how it differs from MCMC (Markov chain Monte Carlo) as mentioned in the article. It would be helpful to include a clear explanation of variational Bayes and its distinctions from MCMC methods.

2. When reading the empirical study, it is difficult to visualize how the model fits the data. As an SEM paper, it is essential to include a path diagram or another visual representation to illustrate the model's fit.

3.  To assist applied researchers, it would be beneficial to provide a more detailed demonstration of the software used, including setup instructions, syntax examples, and potentially screenshots or code snippets to guide users through the process.

4. . The study mentions the importance of checking convergence but fails to explain the convergence diagnosis. Basic steps for implementing Bayesian inference, such as the number of chains, chain length, burn-in, thinning, etc., should be described to ensure a thorough empirical demonstration.

5. I would like to read more discussions on the scenarios where a regularized  SEM should be considered.  It is still unclear why the regularized SEM should be fit to a model, especially since there is only one cross-loading in the empirical study, which is not too complex. 

Author Response

This study reviews the regularized SEM in both frequentist and Bayesian frameworks with the purpose to provide applied users with some training on Bayesian regularized SEM. The study is more like a tutorial paper on how to use regularized SEM. However, it does not provide the necessary details for fitting regularized SEM and for Bayesian inference. I would suggest including more details for future revision.
Thank you for your review and useful suggestions. I have included your suggestions as outlined below.

1. The paper mentions the term ”variational Bayes” without providing an explanation of how it works or how it differs from MCMC (Markov chain Monte Carlo) as mentioned in the article. It would be helpful to include a clear explanation of variational Bayes and its distinctions from MCMC methods.
Thank you for pointing this out. I have added the following paragraph on variational inference and how it differs from MCMC on page 10:
“Whereas MCMC sampling draws directly from the posterior distribution, variational inference approximates the posterior using a simpler distribution. It does so by searching over a family of simple distributions and subsequently finding the distribution that is closest to the posterior according to some criterion.
The advantage over MCMC sampling is that variational inference is typically faster and able to scale better to high-dimensional data sets. The disadvantage is that the approximation might be far from the true posterior distribution.”

2. When reading the empirical study, it is difficult to visualize how the model fits the data. As an SEM paper, it is essential to include a path diagram or another visual representation to illustrate the model’s fit.
A visualization of the model has been included in Figure 3 on page 12.

3. To assist applied researchers, it would be beneficial to provide a more detailed demonstration of the software used, including setup instructions, syntax examples, and potentially screenshots or code snippets to guide users through
the process.
Thank you for this useful suggestion. Apart from the code to rerun the empirical application that was already available online, I have now added a Markdown file (https://github.com/sara-vanerp/BRSsoftware/tree/main/appendix)
summarising the main functions needed to run the analysis with the different packages and refer to this file on page 11.

4. . The study mentions the importance of checking convergence but fails to explain the convergence diagnosis. Basic steps for implementing Bayesian inference, such as the number of chains, chain length, burn-in, thinning, etc., should be described to ensure a thorough empirical demonstration.
I fully agree on the importance of assessing convergence and have therefore added some general guidelines on page 14. How to assess convergence in practice is covered in the Markdown file mentioned in the previous comment. Unfortunately, not all packages offer user-friendly ways to optimally assess convergence so the extent to which these guidelines can be (easily) followed varies between the packages, which is also mentioned.

5. I would like to read more discussions on the scenarios where a regularized SEM should be considered. It is still unclear why the regularized SEM should be fit to a model, especially since there is only one cross-loading in the empirical study, which is not too complex.

Thank you for this suggestion. I have added the following paragraph in the discussion on page 17:
“In general, researchers might consider using regularized SEM whenever their model has many parameters, of which they expect a subset to be equal to zero. Depending on the desired shrinkage behavior, different shrinkage priors can be used. For example, if a large number of substantial parameters is to be expected, it might be needed to use a shrinkage prior that offers non-negligible shrinkage to zero for relevant effects to avoid identification issues. The empirical application considered in this paper has been kept, for illustrative purposes, relatively simple. More advanced applications are possible, although software implementations might be limited.”

Thank you for your comments and I hope you agree these adaptations have sufficiently improved the manuscript for publication

Reviewer 2 Report

The authors propose an interesting review for Bayesian methods for structural equation modeling (SEM). The review is clearly written and exhaustive. An overview of available software packages for regularized SEM is illustrated.
Only few comments:
• Discussion on identification of factors is missing.
• Figure 2 results quite naive in the overall discussion. It would be more interesting a discussion of the effect of prior precision parameters on the posterior shrinkage.
• The robustness on the prior variance is mentioned but a sensitivty analysis on the difference choices would improve the impact of the paper. In literature different priors are chosen [3, 2, 1] and a discussion on the effect on posterior estimates is missing. Line 414 in pag 11 different values for prior variances are mentioned, I would appreciate a mention on the results.

Author Response

The authors propose an interesting review for Bayesian methods for structural equation modeling (SEM). The review is clearly written and exhaustive. An overview of available software packages for regularized SEM is illustrated. Only
few comments:
Thank you for this compliment and please find below the responses to your interesting suggestions.

- Discussion on identification of factors is missing.
I have added the following discussion regarding this point on page 13:
“Note that regularization allows the estimation of parameters that would not be identified in traditional SEM. In this application, we will estimate all crossloadings in addition the the main loadings and factor correlations. Including
a penalty function or shrinkage prior that pulls small estimates sufficiently towards zero ensures identification of the model. However, we still need to impose additional identification constraints to identify the latent variables by either
setting one (main) loading for each latent variable to 1 (unit loading identification) or each latent variable variance to 1 (unit variance identification). By default, lslx and blavaan use unit variance identification by fixing the first main loading of each factor, while LAWBL uses unit variance identification.”

- Figure 2 results quite naive in the overall discussion. It would be more interesting a discussion of the effect of prior precision parameters on the posterior shrinkage.
Thank you for this useful suggestion. I have adapted Figure 2 to show three different normal prior specifications and their subsequent influence on the posterior.

- The robustness on the prior variance is mentioned but a sensitivity analysis on the difference choices would improve the impact of the paper. In literature different priors are chosen and a discussion on the effect on posterior estimates
is missing. Line 414 in pag 11 different values for prior variances are mentioned, I would appreciate a mention on the results.
The original submission included a mention on the results regarding different prior variances. These results are exemplary for all cross-loadings, as is now mentioned explicitly on page 16:
“The results shown in Figure 6 for selected cross-loadings are exemplary for all cross-loadings: as the prior variance decreases and the prior becomes more peaked around zero, it will exert more influence thus shrinking the estimates
more heavily toward zero.”

Thank you for your comments and I hope you agree these adaptations have sufficiently improved the manuscript for publication

Reviewer 3 Report

Author presented an overview on the Bayesian regularized structure equation methods and available software packages. A simulated data set has been used for illustration. The presentation covers wide scopes, and the discussion is well detail. But writing should have major revision. Specially, the abstract should be rewritten completely to address what have been done and delete the unnecessary statements. For example, I do not think the work covers theoretical
development. However, author mentioned “..., with a focus on theoretical developments as well as ...” on the abstract. It would be better that author also addresses how to generate the data set to be used in appendix section. How
many covariate have been involved in the model?

Author Response

Author presented an overview on the Bayesian regularized structure equation methods and available software packages. A simulated data set has been used for illustration. The presentation covers wide scopes, and the discussion is well detail. But writing should have major revision. Specially, the abstract should be rewritten completely to address what have been done and delete the unnecessary statements. For example, I do not think the work covers theoretical
development. However, author mentioned “..., with a focus on theoretical developments as well as ...” on the abstract. It would be better that author also addresses how to generate the data set to be used in appendix section. How
many covariate have been involved in the model?

Thank you for your review and positive words. I have made the following changes based on your comments:
I have changed the sentence “with a focus on theoretical developments as well as available 11 software implementations” in the abstract to: “with a focus on the types of SEMs in which Bayesian regularization has been applied as well as available software implementations”

The model in the empirical application is based on a confirmatory factor model without covariates. The data set that was used is based on an existing data set, but generated to allow freely sharing the data. To generate the data
set, I have used the observed covariance matrix to draw from a multivariate normal distribution with mean vector equal to zero. All the code, including the code to generate the data and reproduce the analyses is available online (see
specifically the file “adapt dataprep.R” for the data generation). To clarify this in the paper, I have adapted the following sentence on page 11:
“A data set with N = 1000 observations was generated by drawing from a multivariate normal distribution with means zero and covariance matrix equal to the complete observed covariance matrix and subsequently splitting into a
training (N = 748) and test (N = 252) set.”

Thank you for your comments and I hope you agree these adaptations have sufficiently improved the manuscript for publication

Reviewer 4 Report

The manuscript entitled ‘Bayesian regularized SEM: Current capabilities and constraints’ gives an overview of current approaches to Bayesian regularized SEM. The authors give a thorough review of the available literature, including use cases and different kinds of approaches to regularization. Furthermore, they illustrate the approach with an empirical example. Overall, the manuscript is well structured and very well written. The introduction and literature review is very informative. The same applies to the overview of the different software packages. I think it makes a modest contribution to the literature. I only have a few comments to further improve the manuscript.

1. The tutorial aspect mentioned in the abstract of the manuscript falls a bit short. In order to increase the comprehensibility of the empirical example, I suggest including relevant code listings directly in the manuscript. I am aware that all code is available at the authors’ github-page, but it facilitates comprehending the actual implementation of the model in the software package(s).

2. Would it be possible to include the Authors’ names in Table 2, instead of only the number referring to the ordering in the reference list? I know it’s the standard referencing in Psych, but in case of the table it would be good to see the names at first glance, instead of having to skip to the reference list.

3. Figures 4 and 6 are hard to read when printing the manuscript in black and white. Would it be possible to increase the dodging between the different estimates or use different shapes?

Minor issues:

M1: On p. 4, line 166 the sentence ‘The high peak’ starts without a space in between.

M2: Table 2 (p. 9) is missing a proper title; ‘This is a wide table’ is just a placeholder.

Author Response

The manuscript entitled ‘Bayesian regularized SEM: Current capabilities and constraints’ gives an overview of current approaches to Bayesian regularized SEM. The authors give a thorough review of the available literature, including use cases and different kinds of approaches to regularization. Furthermore, they illustrate the approach with an empirical example. Overall, the manuscript is well structured and very well written. The introduction and literature review
is very informative. The same applies to the overview of the different software packages. I think it makes a modest contribution to the literature. I only have a few comments to further improve the manuscript.

Thank you for your review and positive words. I have adapted the manuscript based on your useful suggestions, please see below.
1. The tutorial aspect mentioned in the abstract of the manuscript falls a bit short. In order to increase the comprehensibility of the empirical example, I suggest including relevant code listings directly in the manuscript. I am aware that all code is available at the authors’ github-page, but it facilitates comprehending the actual implementation of the model in the software package(s).
Thank you for this useful suggestion. Apart from the code to rerun the empirical application that was already available online, I have now added a Markdown file (https://github.com/sara-vanerp/BRSsoftware/tree/main/appendix)
summarising the main functions needed to run the analysis with the different packages and refer to this file on page 11.

2. Would it be possible to include the Authors’ names in Table 2, instead of only the number referring to the ordering in the reference list? I know it’s the standard referencing in Psych, but in case of the table it would be good to seethe names at first glance, instead of having to skip to the reference list.
I agree that seeing the names at first glance is more convenient and have adapted Table 2 accordingly.

3. Figures 4 and 6 are hard to read when printing the manuscript in black and white. Would it be possible to increase the dodging between the different estimates or use different shapes?
Thank you for pointing this out. Both figures have been adapted to be more readable when printed in black and white.
Minor issues:
M1: On p. 4, line 166 the sentence ‘The high peak’ starts without a space in between.
This has been corrected.
M2: Table 2 (p. 9) is missing a proper title; ‘This is a wide table’ is just a placeholder.
A proper title has been added.
Thank you for your comments and I hope you agree these adaptations have sufficiently improved the manuscript for publication

Round 2

Reviewer 3 Report

Author has revised the paper that is suggested to be published.